# Novel *PRUNE2* Germline Mutations in Aggressive and Benign Parathyroid Neoplasms

**DOI:** 10.3390/cancers15051405

**Published:** 2023-02-23

**Authors:** Sara Storvall, Eeva Ryhänen, Auli Karhu, Camilla Schalin-Jäntti

**Affiliations:** 1Department of Endocrinology, Abdominal Center, University of Helsinki, Helsinki University Hospital, Haartmaninkatu 4, 00290 Helsinki, Finland; 2Department of Applied Tumor Genomics, Research Programs Unit, University of Helsinki, 00290 Helsinki, Finland; 3Department of Medical and Clinical Genetics, University of Helsinki, 00290 Helsinki, Finland

**Keywords:** primary hyperparathyroidism, parathyroid carcinoma, whole-exome sequencing, PRUNE2, germline, mutation

## Abstract

**Simple Summary:**

Recently, *PRUNE2* mutations were indicated in the pathogenesis of aggressive parathyroid neoplasms. Here, we report novel, rare *PRUNE2* mutations located in exons 3, 8, 9, and 12 in patients with parathyroid tumors in the genetically homogenous Finnish population. We identified *PRUNE2* mutations in patients with parathyroid carcinoma, atypical parathyroid tumors, and adenomas. While further research is needed, mutations of the *PRUNE2* gene could play a role in the pathogenesis of parathyroid tumors.

**Abstract:**

Parathyroid tumors are mostly sporadic but can also occur in familial forms, including different kinds of genetic syndromes with varying phenotypes and penetrance. Recently, somatic mutations of the tumor suppressor gene *PRUNE2* were found to be frequent in parathyroid cancer (PC). The germline mutation status of *PRUNE2* was investigated in a large cohort of patients with parathyroid tumors from the genetically homogenous Finnish population, 15 of which had PC, 16 atypical parathyroid tumors (APT), and 6 benign parathyroid adenomas (PA). Mutations in previously established hyperparathyroidism-related genes were screened with a targeted gene panel analysis. Nine *PRUNE2* germline mutations with a minor allele frequency (MAF) of <0.05 were found in our cohort. Five of these were predicted to be potentially damaging and were identified in two patients with PC, two with APT, and three with PA. The mutational status was not associated with the tumor group nor related to the clinical picture or severity of the disease. Still, the frequent finding of rare germline mutations of *PRUNE2* may point to the gene playing a role in the pathogenesis of parathyroid neoplasms.

## 1. Introduction

Parathyroid carcinoma (PC) is a rare cause of primary hyperparathyroidism (PHPT). In Finland, around 2–3 new cases are diagnosed every year [1]. In contrast, PHPT due to benign adenomas is very common with an incidence of 1–2/1000 people [2,3]. The clinical picture of PC is similar to that of PHPT caused by benign parathyroid adenomas (PA). However, PC tumors are often larger than benign adenomas, and the clinical picture is usually more severe. PHPT due to benign adenomas is most common in postmenopausal women (gender ratio 3:1), while for PC, there is no gender difference, and incidence peaks at 40–50 years [4,5]. The diagnosis of PC is set based on unequivocal invasive characteristics on histopathological examination. Surgery is the only possible curative treatment. Of note, so called *en bloc* surgery, with the removal of the ipsilateral thyroid lobe and parathyroid gland gives the patient a better prognosis [6,7]. Atypical parathyroid tumors (APT), also called atypical parathyroid adenomas, are tumors which lack invasive histological characteristics but feature properties such as intratumoral fibrosis and nuclear atypia not regularly found in conventional parathyroid adenomas [8,9]. Around 5% of parathyroid adenomas (up to 15%, according to some studies) can be classified as APT [2,9,10].

The vast majority of PC cases are sporadic (80–90%) and associated with somatic mutations, although PC can also be hereditary [8]. Familial PHPT can occur as isolated PHPT or syndromes, presenting with both benign parathyroid adenomas as well as PC. Germline inactivation of the *CDC73* gene, coding for the protein parafibromin, is associated with hyperparathyroidism jaw–tumor syndrome (HPT-JT). The parafibromin protein functions as a tumor suppressor, and HPT-JT presents with early-onset PHPT, ossifying jaw fibromas, as well as tumors of the kidney and uterus. However, *CDC73* mutation carriers display incomplete penetrance, and germline *CDC73* mutations can also cause isolated familial PHPT without the HPT-JT phenotype [4,11]. Up to 30% of PC patients carry inactivating germline *CDC73* mutations [9]. Due to incomplete penetrance, these PC patients do not always present with a family history of PHPT. Somatic inactivation of *CDC73* in tumor DNA is common in sporadic PC, occurring in up to 75% of cases [12,13].

Parathyroid neoplasms can also be associated with multiple endocrine neoplasia (MEN) syndromes MEN1, MEN2A, and MEN4, caused by mutations in the genes *MEN-1*, *RET,* and *CDKN1B*, respectively [8,14,15,16]. While loss-of-function mutations in calcium-signaling-related genes, such as *CASR*, *GNA11*, and *AP2S1*, are associated with familial hypocalciuric hypercalcemia (FHH) and neonatal severe hyperparathyroidism (NSHPT), these mutations are not associated with parathyroid neoplasms [17,18,19,20].

Non-syndromic hereditary hyperparathyroidism, called familial isolated hyperparathyroidism (FIHP), can be caused by the incomplete manifestation of mutations of syndrome-related genes such as *MEN-1*, *CDC73*, *GCM2*, *CDKN1A*, *CDKN1B,* or *CDKN2C* [17,21]. However, up to 70% of FIHP cases lack mutations in known susceptibility genes, indicating that there are many factors not yet discovered in the pathogenesis of parathyroid tumors [17,21]. Additional germline mutations reported in PC patients include genes *PRUNE2*, *CCD1*, *ADCK1,* and genes of the PI3K/AKT/mTOR pathway [22,23,24,25].

*Prune homolog 2 with BCH domain* (*PRUNE2*, also known as *BMCC1*) (9q21.2, ENSG00000106772) belongs to the B-cell CLL/lymphoma 2 and adenovirus E1B 19kDa interacting family. Members of this gene family are involved in several cellular processes, such as apoptosis, cell transformation, and synaptic function [26,27]. *PRUNE2* is a tumor suppressor gene of particular significance shown in prostate cancer, where its expression is regulated by *prostate cancer antigen 3* (*PCA3*), a non-protein coding gene located on the opposite DNA strand in an intron of *PRUNE2* [28]. *PCA3* has been investigated as a biomarker in prostate cancer [29,30]. Increased *PCA3* levels have also been found in other cancers such as choriocarcinoma as well as ovarian and thyroid cancer, indicating involvement of *PRUNE2* in the pathogenesis of these cancers [31]. Increased PRUNE2 protein levels are associated with a favorable prognosis in neuroblastoma and leiomyosarcoma and have also been associated with lower tumorigenic activity in colorectal cancer [26,27]. Somatic *PRUNE2* mutations have been reported in up to 18% of sporadic PCs. These mutations, in combination with the loss of heterozygosity (LOH) events in the tumors, suggest *PRUNE2* involvement in the pathogenesis of PC [24,25]. Moreover, Yu et al. identified a rare germline missense mutation in one PC patient without other known PC-driving mutations. The remaining wild-type allele was silenced in the patient’s tumor by LOH, a feature characteristic of tumor suppressor genes. However, the role of *PRUNE2* in the genetic predisposition of PC is yet to be confirmed [25].

As somatic *PRUNE2* mutations are considered significant in the pathogenesis of PC, we wanted to study whether germline *PRUNE2* mutations might be identified in patients diagnosed with aggressive parathyroid tumors in the genetically homogenous Finnish population, an ideal population for the discovery of rare genetic defects of monogenic diseases [32].

## 2. Materials and Methods

Our study cohort consists of 37 patients with PHPT, 15 of which have been diagnosed with PC, 16 with APT, and 6 with PA. Three patients are known to have an ethnicity other than Finnish. All patients have been diagnosed and treated for PHPT during the years 2004–2021 at the Endocrine Department of Helsinki University Hospital. Clinical history, surgical and pathological reports as well as laboratory results at diagnosis were collected from the Helsinki University Central Hospital databases. Patients with previously known *CDC73* mutations were reported earlier and are not included in this study [11].

PC patient 152164 is the mother of APT patient 152177; otherwise, the patients are not related to each other and do not have known family history of PHPT. All patients were summoned for laboratory tests. Serum ionized calcium (S-Ca-ion) as well as fasting plasma parathyroid hormone (fP-PTH), serum vitamin D (S-D-25-OH), and serum creatinine were also analyzed to assess possible recurrence of primary hyperparathyroidism, as some patients had not been participating in regular follow-ups. For some patients, due to problems with sample delivery, DNA was not extracted from whole blood but from saliva samples sent to the patients’ home addresses, using the Ora-Collect OCR-100 sample collection kit (Ottawa, Ont., Canada)).

All patients gave their written informed consent to study participation. The study protocol was approved by the Ethics Committee of the Helsinki University Hospital (Dnro 1803/2018).

EDTA whole-blood samples were sent to Blueprint Genetics (www.blueprintgenetics.com, accessed on 3 January 2023, Helsinki, Finland)) for DNA extraction and whole exome sequencing (WES). WES was performed using BpG according to in-house methods and sequence alignment to human reference genome GRCh37/hg19 with Illumina’s bcl2fastq2 Software v2.20 (Illumina, Inc. San Diego, CA, USA).

Mutations in previously known PHPT-related genes were analyzed and reported according to a customized extended BpG hyperparathyroidism gene panel (www.blueprintgenetics.com, accessed on 3 January 2023, Helsinki, Finland, including genes *AIP*, *AIRE*, *AKAP9*, *AP2S1*, *APC*, *BRCA2*, *CASR*, *CDC73*, *CDKN1A*, *CDKN1B*, *CDKN2B*, *CDKN2C*, *CEP152*, *CTNNB1*, *DICER1*, *EZH2*, *FLNA*, *GCM2*, *GNA11*, *KDM5C*, *MEN1*, *MTOR*, *NF1*, *NTRK1*, *PIK3CA*, *PTEN*, *PTH*, *RB1*, *RET*, *SDHA*, *SETD1B*, *SMARCA4*, *STK11*, *TERT*, *TNRC6A*, *TP53*, *TRPV6*, *TSC1*, *TSC2*, *WT1*, and *ZEB1.*

The gnomAD38 v.2.1.1 database (https://gnomad.broadinstitute.org/(accessed on 3 January 2023)) was used as a resource for population frequencies of the germline mutations. Previously known mutation consequences were assessed through the ClinVar database (https://www.ncbi.nlm.nih.gov/clinvar/(accessed on 3 January 2023)) [33]. For assessing the pathogenicity of the variants PolyPhen-239 v.2.2.3 (http://genetics.bwh.harvard.edu/pph2/(accessed on 4 January 2023)) and SIFT v5.1.1 (http://sift-dna.org (accessed on 4 January 2023)) mutation consequence predictors were used. PhyloP conservation scores were gathered from the UCSC database (http://genome.ucsc.edu (PhyloP version 3.19, accessed on 4 January 2023)) [34]. Protein sequence and structural information about the *PRUNE2* gene was gathered from the Ensembl database (v.108) [35].

Visualization of *PRUNE2* single nucleotide variants and small insertions/deletions were performed with the in-house developed analysis and visualization program BasePlayer [36]. Minimum coverage for variant calling was set at four reads, and the mutated allele was required to be present in at least 20% of the reads. Output processing and analysis was performed using RStudio v.1.1.463 (RStudio, Inc., Boston, MA, USA) and IBM SPSS Statistics v.27 (SPSS, Inc., Chicago, IL, USA). *p*-values of <0.05 (two-tailed) were considered statistically significant. The χ^2^-test with Fisher’s exact test, as appropriate, was used to investigate differences in categorical variables between groups, while the Kruskal–Wallis test was used for continuous variables. For *PRUNE2* mutations, those with a frequency of less than 5% in the global and/or Finnish population are reported. Genome position is reported according to *PRUNE2* transcript ENST00000376718.

## 3. Results

### 3.1. Overview of the Main Study Results and Patient Characteristics

A flowchart depicting the study design and the main findings is shown in Figure 1.

The patient characteristics are shown in Table 1.

### 3.2. Hyperparathyroidism Gene Panel Findings

Gene panel findings in the cohort are listed in Table 2.

The BpG custom hyperparathyroidism gene panel (www.blueprintgenetics.com, accessed on 3 January 2023, Helsinki, Finland did not reveal any additional germline *CDC73* mutations in our cohort. One PC patient (152164) is the mother of one of the APT patients (152177). Otherwise, the patients are not related to each other and do not have any known cases of hyperparathyroidism in their respective families. Three PC patients are known to have an ethnicity other than Finnish. All patients were alive at follow-up. More than one surgery was performed on nine PC patients; five patients had local or distant PC recurrence. Additional *en bloc* surgery was performed as a preventative measure on four PC patients, as *en bloc* was not performed as the primary surgery. Three APT patients had more than one surgery; in all these cases, additional surgery was performed due to persistent hypercalcemia after the initial surgery, or the patient presented with additional parathyroid pathology (adenoma or hyperplasia). None of these patients were carriers of any potentially damaging germline mutations found in this study.

One PC patient (patient ID 152172) was found to have a likely pathogenic heterozygous germline *MEN-1* mutation (c.1280G > T, p.Ser427Ile, ENST00000312049.6), although displayed no other clinical features of the MEN1 syndrome (Figure 1, Table 2) [37,38]. One APT patient had a family history of MEN1, but no pathogenic *MEN-1* variants were detected in this patient. The custom WES panel revealed monosomy X (Turner syndrome) in one APT patient (patient ID 152142) as an incidental or secondary finding (ISF) with clinical significance (Figure 1). Moreover, a heterozygous germline *APC* missense mutation (c.2222A > G, p.Asn741Ser, rs150209825) was found in PA patient 152129. The variant is considered of unknown significance (VUS) [39,40,41]. This patient does not have any history of other neoplasia, and the possible family history of cancer is unknown. A 5′UTR variant of the *APC* gene (c.-128G > A, rs543098847) was found in PC patient 152174. This multiallelic single nucleotide variant is considered likely benign VUS (https://www.ncbi.nlm.nih.gov/clinvar (accessed on 3 January 2023)). A *RET* mutation (c.604G > A, p.Val202Met, rs751572082) of unknown significance was identified in PC patient 152165. An amino acid deletion of *BRCA2* with unknown significance (c.3900_3902del, p.Met1300_Thr1301delinsIle, rs397507697) was found in APT patient 152175, and APT patient 152146 had a VUS missense *AIP* mutation (c.940C > T, p.Arg314Trp, rs375740557). In PA patient 152125, a multiallelic 5′UTR *SDHA* variant (c.-11C > T, rs1396057630) was identified.

### 3.3. PRUNE2 Mutations

Altogether, 25 non-synonymous *PRUNE2* germline variants were found in the patient cohort. Out of those, nine variants had a minor allele frequency (MAF) <0.05 in a control population (https://gnomad.broadinstitute.org (accessed on 21 December 2022)) (Figure 1). The details on these nine *PRUNE2* variants are listed in Table 3.

The nine variants were distributed among twelve patients, of which three had PC, six APT and three PA. None of the patients with rare *PRUNE2* variants had mutations of previously established PHPT-related genes (Table 2). All discovered mutations were heterozygous missense changes. Altogether, seven patients harbored a *PRUNE2* variant in silico predicted to be likely deleterious (Figure 1, Table 3). Interestingly, the germline variant p.Ser595Tyr found in APT patient 152142 was previously also identified in prostate cancer patients [42]. Otherwise, the rare *PRUNE2* germline variants found in our cohort have not been associated with any kind of pathology (https://www.ncbi.nlm.nih.gov/clinvar/(accessed on 3 January 2023)). APT patient 152142 with Turner syndrome, who was 59 years old at the time of her diagnosis, had three different germline *PRUNE2* variants; however, only (c.1784G > T, p.Ser595Tyr) was predicted to be damaging (Table 3). More than one *PRUNE2* mutation was also found in patients 152127 (PA, 38 years old at diagnosis) and 152171 (PC, 66 years at diagnosis). Similarly, in these cases, only one mutation per patient was predicted to be pathogenic. In addition, patients 152155 (PC, 49 years at diagnosis), 152121 (PA, 44 years at diagnosis), 152123 (PA, 54 years at diagnosis), and 152131 (APT, 63 years at diagnosis) each harbored one *PRUNE2* germline mutation that was predicted to be damaging (Table 3). The location of the *PRUNE2* mutations found in our cohort in relation to the *PRUNE2* gene and previously found *PRUNE2* mutations in PC (somatic and germline) are visualized in Figure 2.

Most mutations were located in exon 8, the longest exon of the gene, but mutations were also observed in exons 3, 9, and 12. The cohort was screened for gene variants of *ADCK1*, *CCD1*, *FAT3*, and *THRAP3* that have previously been associated with PC as described in the literature but that were not included in the BpG hyperparathyroidism gene panel assessment [4,16,25,27]. No previously described variants of these genes were found in our cohort.

*PRUNE2* mutation status did not correlate with clinical parameters such as severity of disease (Ca-ion or PTH levels at diagnosis), tumor size, or parafibromin staining on immunohistochemistry, neither by looking at all rare prune mutations in our cohort nor by separately analyzing the variants considered damaging. Neither were the *PRUNE2* mutations associated with a lower age at diagnosis.

## 4. Discussion

The majority of PC cases are associated with somatic alterations, although hereditary predisposition also plays a role in the genesis of this rare malignancy [14,43]. Somatic recurrent *PRUNE2* mutations have been reported in up to 18% of PC cases. A reported germline *PRUNE2* mutation and inactivation of the wild-type allele by LOH in the patient’s tumor indicates that *PRUNE2* alterations might predispose to PC [24,25]. To clarify the contribution of inherited *PRUNE2* mutations in the development of parathyroid tumors, the germline mutation status of *PRUNE2* was analyzed in a cohort of 37 mostly Finnish PHPT patients. We identified nine rare *PRUNE2* germline variants (MAF < 0.05) in twelve patients, of which five mutations were predicted to be deleterious by disrupting the function of the *PRUNE2* protein. These five mutations were distributed among seven patients, including individuals from all tumor groups (PC, APT, and PA). Interestingly, one of the mutations (c.270C > T, p.Asp90ASn) was shared between three unrelated patients: two with PC and one with PA. The MAF of the variant was 0.0148 among the Finnish controls and 0.0092 in the global control population (Table 3). This recurrent mutation might be a Finnish founder mutation, and enrichment of the mutant allele in our patient cohort may imply a causal relationship between the c.270C > T (p.Asp90ASn) mutation and the disease phenotype. The homogenous Finnish population has a unique genetic background, and founder mutations exist at high frequencies [44]. As such, Finnish population-based cohorts are valuable for the discovery of the causative genetic mutations of monogenic diseases.

Most of the rare PRUNE2 variants found in our cohort are localized to PRUNE2 exon 8, which is, by far, the largest exon of the gene, harboring approximately half of all the amino acids in its sequence. The mutations found in our study do not seem to be targeting any specific regions or previously established domains of PRUNE2 [45].

According to the literature, germline mutations of the *PRUNE2* gene have been very scarcely investigated in other forms of cancers, despite somatic mutations playing a role in the pathogenesis of prostate cancer, leiomyosarcoma, and colorectal cancer among others [26,27,29,30,31]. Of note, a recent study found *PRUNE2* germline mutations in 2.8% of patients with familial prostate cancer, proposing *PRUNE2* as a new prostate cancer predisposition gene [42]. The *PRUNE2* germline variant c. 1784G > T (p.Ser595Tyr) found in our APT patient was also identified in the familial prostate cancer study. This mutant allele was classified as a variant of unclear association with the prostate cancer risk [42].

The patients with rare *PRUNE2* germline mutations did not carry any germline mutations in previously established PHPT-driving genes. One patient was found to have monosomy X/Turner syndrome. This is relevant in the setting of parathyroid tumors due to the X chromosome harboring loci for the *FLNA* and *KDM5C* genes. The FLNA protein participates in the regulation of the calcium-sensing receptor and has been associated with increased aggressiveness in a wide range of cancers [46,47,48]. FLNA expression has also been investigated in parathyroid tumors, with unclear conclusions [49,50]. Inactivation of the *KDM5C* gene encoding for the histone demethylase protein JARID1C is frequent in renal cell carcinoma, and somatic *KDM5C* mutations have also been found in PC [25,51]. The monosomy X patient is thus susceptible to somatic mutations in these genes. One patient (152172) was discovered to have a likely pathogenic mutation of the *MEN-1* gene. Other mutations of the same codon (c.1281T > A, p.Ser427Arg, rs1114167528) were previously reported in several MEN1 patients [37,38]. The family history of the *MEN-1* mutation-positive patient identified in this study is not known. The patient has no other medical history of cancer except for PC, and the age at diagnosis (67 years) was not conspicuous. Still, due to the likely pathogenicity of this gene mutation, further clinical follow-up might be needed. Similarly, despite the germline *APC* mutations discovered in our cohort not being previously reported to be associated with pathogenicity, these patients and their relatives might also require further investigations, as *APC* mutations are so pronouncedly associated with hereditary colorectal malignancy [41,52].

Patients with previously known *CDC73* mutations were excluded from this study, but as germline *CDC73* mutations are also quite common in patients with sporadic PC it is perhaps rather surprising that none of the patients in the present cohort were found to carry *CDC73* mutations. As *CDC73* mutations may underlie both benign and malignant parathyroid tumors with varying penetrance and phenotypes, one can, therefore, speculate that mutations of the *PRUNE2* gene could also give rise to similarly varying parathyroid tumor phenotypes with similarly varying penetrance [6,53,54]. The incomplete penetrance manifested by *PRUNE2* germline defects might be the reason for the observed lack of family history of the disease.

The shortcoming of the study was the lack of tumor material; hence, the investigation of the biallelic inactivation of *PRUNE2* in tumors was not achievable. Our patient cohort is of a reasonable size considering the rarity of PC. Another strength of our material is the detailed clinical, surgical, and histopathological characterization of the patients. However, the patient number is too small for the relevant assessment of the possible relationships between gene variants and clinical or biochemical parameters. All our patients were alive at the time of the study, with a median follow-up time of 7 years. Globally, the 5-year survival of PC is considered around 85%, indicating that our patients have had a rather favorable course of disease [1,55,56]. En bloc surgery was performed in 13 of our 15 PC patients (86%), either as primary or secondary surgery. The excellent prognosis of the patients likely reflects the high awareness of PC as a cause of PHPT in our tertiary centre and the close collaboration with our endocrine surgeons, preventing diagnostic delay, as well as en bloc surgery, ensuring margin-free resection. Recently, margin-free resection was reported to predict excellent long-term outcomes in PC [7].

In this study, we report that Finnish PHPT patients with *CDC73* mutation-negative parathyroid tumors frequently display rare germline mutations in the *PRUNE2* tumor suppressor gene. However, further work is needed to examine whether *PRUNE2* plays a causative role in the genetic predisposition of parathyroid neoplasia. Clarification of this question would require additional sample sets and more extensive molecular workup. The identification of new parathyroid tumor-predisposing genes is important to improve the risk assessment of patients, and it would enable targeted testing of family members at risk.

## 5. Conclusions

Rare germline *PRUNE2* variants are frequent in Finnish patients with parathyroid neoplasms, regardless of tumor type (PC, APT, or PA). Further studies are needed to clarify the role of *PRUNE2* in patients with parathyroid tumors.

## Figures and Tables

**Figure 1 cancers-15-01405-f001:**
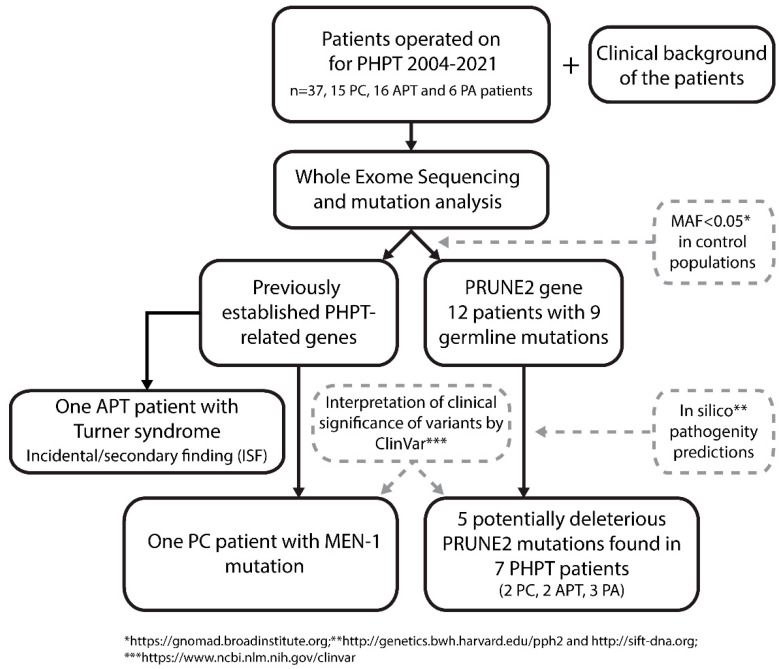
Flowchart of *PRUNE2* germline mutation screening in patients with parathyroid tumors. The whole exome sequencing (WES) was performed from normal DNA. After WES analysis, the variant calls were filtered to have MAF < 0.05. Mutation consequences were assessed using pathogenicity predictor software and the ClinVar database. Five potentially deleterious *PRUNE2* mutations were identified, distributed among seven PHPT patients. PC = parathyroid cancer; APT = atypical parathyroid tumor; PA = parathyroid adenoma; MAF = minor allele frequency.

**Figure 2 cancers-15-01405-f002:**
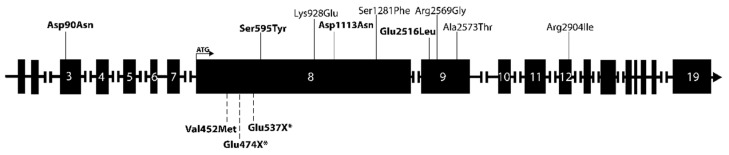
Schematic depiction of the *PRUNE2* gene and its exons. *PRUNE2* variants found in our cohort are annotated above the gene, and previously identified *PRUNE2* mutations (somatic and germline) in PC patients are annotated below the gene. The respective sizes of the exons are not to scale. Somatic mutations are marked with an asterisk (*). Mutations predicted to be damaging are marked in bold.

**Table 1 cancers-15-01405-t001:** Characteristics of the patient cohort.

	PC	APT	PA	Total	*p*-Value
*n*.	15	16	6	37	
Sex (m:f)	8:7	5:11	0:6	13:25	0.054
Age at diagnosis (years, median, range)	61 (17–76)	60 (33–80)	58 (38–73)	60 (17–80)	
S-Ca-ion at diagnosis (mmol/L, median, range)	2.05 (1.62–2.58)	1.73 (1.32–2.39)	1.50 (1.41–1.64)	1.8 (1.32–2.58)	<0.001 *
S-PTH at diagnosis ng/L (median, range)	1743 (358–4000)	330 (73–3500)	138 (63–222)	502 (63–4000)	<0.001 *
More than one surgery for PHPT (*n*. of patients)	9	3	0	8	0.271
Hypercalcemic crisis or hypercalcemia requiring in-hospital treatment	10	6	0	16	0.015 *
Palpable neck mass	2	2	0	4	0.689
Renal failure (transient or permanent serum creatinine elevation)	5	4	0	9	0.320
Skeletal manifestation (osteoporosis or osteitis fibrosa cystica)	4	6	1	11	0.713
Number of patients with rare PRUNE2 mutations	3	7	3	13	0.343

*p*-values marked with an asterisk (*) are considered statistically significant. PC; parathyroid carcinoma; PA: Parathyroid adenoma; APT: atypical parathyroid tumor; PHPT: primary hyperparathyroidism.

**Table 2 cancers-15-01405-t002:** Results of the customized Blueprint Genetics (BpG) hyperparathyroidism gene panel analysis.

Patient ID	Diagnosis	Age at Diagnosis	Ethnicity (If Other Than Finnish)	Mutated Gene	Mutation	Predicted Consequence
152155	PC	49				
152161	PC	61	Russian			
152163	PC	45				
152164	PC	53				
152165	PC	35		*RET*	c.604G > A p.Val202Met	VUS
152168	PC	71				
152170	PC	71				
152171	PC	66				
152172	PC	67	Estonian	*MEN1*	c.1280G > T, p.Ser427Ile	Likely pathogenic
152174	PC	65		*APC*	5′UTR variant c.-128G > A	VUS
156889	PC	76				
167873	PC	40				
167874	PC	17				
171375	PC	46	Russian			
177360	PC	72				
152131	APT	63				
152133	APT	80				
152134	APT	52				
152135	APT	56				
152136	APT	68				
152137	APT	60				
152140	APT	70				
152142	APT	59		Monosomy X (Turner syndrome)		
152143	APT	50				
152146	APT	52		*AIP*	c.940C > T, p.Arg314Trp	VUS
152149	APT	67				
152169	APT	56				
152175	APT	70		*BRCA2*	Inframe deletion c.3900_3902del p.Met1300_Thr1301delinsIle	VUS
152176	APT	64				
152177	APT	33				
177359	APT	49				
152121	PA	44				
152123	PA	54				
152125	PA	73		*SDHA*	5′UTR variant c.-11C > T	VUS
152127	PA	38				
152128	PA	68				
152129	PA	61		*APC*	c.2222A > G p.Asn741Ser	VUS

**Table 3 cancers-15-01405-t003:** Identified rare *PRUNE2* germline mutations (MAF < 0.05) according to underlying tumor in 37 patients with primary hyperparathyroidism.

Patient ID	Dg (Age at Onset/Sex)	*PRUNE2* Mutation (Exon)	* MAF (Finnish)	* MAF (Global)	Predicted Effect SIFT /Polyphen	Phylop Conservation Score	rs Number
152155	PC (49/F)	c.270C > T, Asp90Asn (3)	0.01477	0.009245	Deleterious/probably damaging	5.56559	rs41304230
152164	PC (53/F)	c.7719C > T Ala2573Thr (9)	0.01656	0.02567	Tolerated/benign	0.280654	rs56261747
152171	PC (66/M)	c.270C > T, Asp90Asn (3)	0.01477	0.009245	Deleterious/probably damaging	5.56559	rs41304230
		c.3842G > A, Ser1281Phe (8)	0.01263	0.01024	Tolerated/possibly damaging	2.67864	rs41310047
152131	APT (63/M)	c.2547TC > AA ^+^, p.Glu2516Leu (9)	0.001639, 0.001641	0.001639, 0.001641	Deleterious/probably damaging	0.104701, 0.36863	rs187947807, rs190606277
152134	APT (52/F)	c.2784T > C, Lys928Glu (8)	0.03	0.03	Tolerated/benign	0.747535	rs41289953
152140	APT (70/F)	c.2784T > C, Lys928Glu (8)	0.03	0.03	Tolerated/benign	0.747535	rs41289953
152142	APT (59/F)	c.1784G > T, Ser595Tyr (8)	0.01447	0.001681	Deleterious/probably damaging	2.36166	rs201792781
		c.7707T > C, Arg2569Gly (9)	0.01659	0.02568	Tolerated/benign	1.07244	rs41288767
		c.7719C > T, Ala2573Thr (9)	0.01656	0.02567	Tolerated/benign	0.280654	rs56261747
152149	APT (67/F)	c.2784T > C, Lys928Glu (8)	0.03	0.03	Tolerated/benign	0.747535	rs41289953
152176	APT (64/F)	c.2784T > C, Lys928Glu (8)	0.03	0.03	Tolerated/benign	0.747535	rs41289953
152121	PA (44/F)	c.3339C > T, Asp1113Asn (8)	0.004178	0.0032776	Deleterious/possibly damaging	4.12181	rs200875180
152123	PA (54/F)	c.270C > T, Asp90Asn (3)	0.01477	0.009245	Deleterious/probably damaging	5.56559	rs41304230
152127	PA (38/F)	c.8711C > A, Arg2904Ile (12)	0.0162	0.003269	Deleterious/probably damaging	1.7718	rs80290481
		c.7719C > T, Ala2573Thr (9)	0.01656	0.02567	Tolerated/benign	0.280654	rs56261747
		c.7707T > C, Arg2569Gly (9)	0.01659	0.02568	Tolerated/benign	1.07244	rs41288767

* https://gnomad.broadinstitute.org (accessed on 4 January 2023); ^+^ combined missense GAa > TTa variant. PhyloP: https://genome-euro.ucsc.edu/(version 3.19, accessed on 4 January 2023) (GRCh37/Hg19). MAF: mutant allele frequency; PC; parathyroid carcinoma; PA: parathyroid adenoma; APT: atypical parathyroid tumor.

## Data Availability

The data is available on request from the corresponding author.

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
