# Peer review of "Novel PRUNE2 Germline Mutations in Aggressive and Benign Parathyroid Neoplasms"

_cancers, 2023, doi:10.3390/cancers15051405_

Round 1

Reviewer 1 Report

This article centers on how PRUNE2 is mutated quite abundantly in your parathyroid cancer cohort. Albeit a small cohort, it is an interesting finding that could be further looked at in a future study with greater numbers in order to robustly determine whether or not PRUNE2 mutations are drivers for PC. I find the MFA analysis to be a neat addition, and would enjoy if there were more data, or at least a figure the authors could include to illustrate to readers how it may be a founder mutation that your screen picked up upon.

Major Edits: Table 2 and Table 3 are not formatted in a way that they are clear on a pdf. Otherwise there are no issues with the figures (except that an additional figure for MFA, and perhaps a survival w/ and w/o the PRUNE2 mutations (somewhat described in lines 295-297) would enhance the paper's impact and readability.

Minor Edits: Use of the Oxford comma throughout (add a "," in lines 10 & 97).

In lines 56-59- Are these numbers correct? If '80% of PC are sporatic', and if 75% of these have CDC73 mutations, how is the total PC CDC73 mutational burden  only ~30%? It would seem like the % would be greater.

In lines 60-67 you list a number of PC related genes, but then state that in familial PC that a majority of the cases DO NOT detect any of these commonly associated genes as being mutated. Does this factor into your PRUNE2 observations at all? In the sense that your PRUNE2 mutated cohort did not have other PC associated driver gene mutations.

Line 96: 'Our study consisted of 37 patients' instead of material consisted of 37 patients. Additionally, here would be better to include your exclusion criteria discussed in the Discussion (line 280).

Reviewer 2 Report

Congratulations to the authors who carried out the work.

This is an interesting paper regarding Parathyroid carcinomas.

Even obtaining samples of PC patients is a process that can take a very long time. Evaluation of the possible association of rare PC patients with a specific mutation in a particular population will contribute to cancer etiology. Minor comments;

*Databases (gnomAD, PolyPhen-2, Clinvar, etc.) are effectively used and presented in the study. The clinical features of the patients are well expressed.

*Please make the resolution of Figure 1 better.

*Recurrent surgery indication was observed due to clinically recurrent PHPT history. Is there a different significance for this specific group for the PRUNE2 mutation? such as the severity of the mutation they carry.

Reviewer 3 Report

This paper is a report of germline mutations in 7 patients with parathyroid tumors. Although somatic mutations of the prune 2 gene in parathyroid tumor tissue have already been reported, germline mutations of the prune 2 gene in patients with parathyroid tumors have been reported in only one case. In that sense, I consider this paper to be significan.

1.    Considering that three of the seven cases with germline mutation of prune 2 found in this study were adenomas, these parathyroid tumors are not necessarily aggressive parathyroid neoplasms. Therefore, the word aggressive in the title should be changed.

2.    The basis for the presumption of germline mutation in this study is described in Materials and Methods, but this alone is not sufficient to determine that it is a germline mutation. It is necessary to investigate the relationship between segregation and phenotype in the family relatives of these cases.

Reviewer 4 Report

I have read with great interest the manuscript entitled "Novel PRUNE2 germline mutations in aggressive parathyroid 2 neoplasms”. In this study the authors used Finnish patients of which 15 had parathyroid carcinoma, 16 atypical parathyroid tumors, and 6 benign adenomas in search for genetic mutations. PRUNE2 mutations were found in 13 patients.

This is a well-written manuscript and the results are very nicely presented including the data in tables and the authors should be congratulated on their interesting work.

I have no comments.
